# Resistance towards ChadOx1 nCoV-19 in an 83 Years Old Woman Experiencing Vaccine Induced Thrombosis with Thrombocytopenia Syndrome

**DOI:** 10.3390/vaccines10122056

**Published:** 2022-11-30

**Authors:** Constant Gillot, Julien Favresse, Vincent Maloteau, Valérie Mathieux, Jean-Michel Dogné, François Mullier, Jonathan Douxfils

**Affiliations:** 1Department of Pharmacy, Namur Research Institute for Life Sciences, Namur Thrombosis and Hemostasis Center, University of Namur, B-5000 Namur, Belgium; 2Department of Laboratory Medicine, Clinique St-Luc Bouge, B-5000 Namur, Belgium; 3Service d’Hématologie, CHU UCL NAMUR-Site Sainte Elisabeth, Namur Thrombosis and Hemostasis Center, B-5000 Namur, Belgium; 4Université Catholique de Louvain, CHU UCL NAMUR, Department of Laboratory Medicine, B-5300 Yvoir, Belgium; 5Qualiblood sa, Research and Development Department, B-5000 Namur, Belgium

**Keywords:** resistance, vaccine, adenovirus, SARS-CoV-2, diagnostic

## Abstract

Background: in this report, we describe the case of an 83-year-old woman vaccinated with ChadOx1 nCoV-19 who developed a so-called vaccine-induced thrombosis with thrombocytopenia syndrome and who did not develop any antibodies against the spike protein of SARS-CoV-2 at 30 days following the administration of her first dose of ChadOx1 nCoV-19. Experimental section: two serum samples from the patient and 5 serum samples from 5 control individuals having received the two-dose regimen vaccination with ChadOx1 nCoV-19 were evaluated. In order to investigate the lack of response to the vaccination, a cell model was developed. This model permits to evaluate the interaction between responsive cells (A549) possessing the Coxsackievirus and Adenovirus Receptor (CAR), a defined concentration of ChadOx1 nCoV-19 and serial dilution of the patient or the control serum. The aim was to assess the impact of these sera on the production of the spike (S) protein induced by the transfection of the genetic material of ChadOx1 nCoV-19 into the A549 cells. The S protein is measured in the supernatant using an ELISA technique. Results: interestingly, the serum from the patient who developed the vaccine-induced thrombosis with thrombocytopenia syndrome impaired the production of S protein by the A549 cells transfected with ChadOx1 nCoV-19. This was not observed with the controls who did not interfere with the transfection of ChadOx1 nCoV-19 into A549 cells since the S protein is retrieved in the supernatant fraction. Conclusion: based on the data coming from the clinical and the cell model information, we found a possible explanation on the absence of antibody response in our patient. She has, or has developed, characteristics that prevent the production of the S protein in contrast to control subjects. We were not able to investigate the entire mechanism behind this resistance which deserve further investigations. A link between this resistance and the development of the thrombosis with thrombocytopenia syndrome following vaccination with ChadOx1 nCoV-19 cannot be excluded.

## 1. Introduction

The use of adenoviruses to develop vaccination strategies has been known for almost 20 years [1]. The use of these viral vectors is ideal for vaccine therapy due to their ability to induce innate immunity and adaptive immunity in the host [1] adenoviruses generate a strong immune response. The pathway they are assumed to use to activate the immune system is certainly one involving the expression of so-called pathogen-associated molecular patterns (PAMPs). PAMPs bind to pathogen recognition receptors on the cells of the host, including those of the innate immune system, thereby initiating the production of proinflammatory cytokines and the differentiation of immature dendritic cells into professional antigen-presenting cells [2]. In addition, adenoviruses infect cells and release the gene product encoding the antigenic proteins that enable their production inside the host cell [3]. While human adenoviruses are the non-replicating viral vectors most commonly used in the development of the SARS-CoV-2 vaccine, pre-existing immunity to the viral vector may interfere with the induction of immunity, particularly after repeated doses [3]. Pre-existing immunity may reduce the capacity of the virus to enter the human cell, reducing the ability of the vaccine to induce a strong immune response. Such phenomenon is more frequently described with human adenoviruses, an example being demonstrated by the failure of the Ad5-nCoV (CanSino^®^, CanSino Biological Inc., Beijing Institute of Biotechnology, Beijing, China) to generate high antibody response in the subjects which had pre-existing antibodies against Ad5 [4]. Pre-existing humoral immunity is correlated to poor transgene expression and a diminished CD8 T cell response against the desired antigen [5,6,7]. As the majority of anti-Ad5 neutralizing Abs (NAbs) are directed against the major coat protein hexon, more specifically its solvent-exposed hypervariable regions (HVRs), and as this hexon is not involved in the coxsackie and adenovirus receptor (CAR)-dependent entry mechanism that Ad5 uses, the neutralizing effect is unlikely to involve classical entry blocking [8,9]. Recent investigations postulates that the cytosolic receptor TRIM21 could be involved via a mechanism that is still not well understood [10].

In order to overcome this risk, chimpanzee adenovirus modified to contain a fragment of the genetic material of the targeted pathogen has been proposed as alternative to these human adenoviruses [11,12]. ChadOx1 nCoV-19 is an adenovirus-based vaccine against SARS-CoV-2 developed by Astrazeneca (Cambridge, UK), in collaboration with the Oxford University. It is distributed worldwide under the brand name of Vaxzevria^®^ (Astrazeneca, Cambridge, UK) [11].

Once infected by the adenovirus, human cells produce the SARS-CoV-2 spike protein (S) and secrete it outside the cells to trigger an immune response [11,12]. The secreted spike protein is detected by antigen presenting cells which, after the recruitment of T-helper cells and the subsequent stimulation of B lymphocytes, stimulate the production of antibodies directed against the spike protein [13]. The spike protein is responsible for the adhesion and the fusion of SARS-CoV-2 with the cell host and therefore antibodies which can target the spike protein have the potential to neutralize the entry of the virus into human cells [3]. Importantly, only cells presenting at their surface the receptors needed for the entry of the adenovirus are potentially able to integrate the DNA of SARS-CoV-2 into their cytoplasm. In the case of ChadOx1 nCoV-19, cells need to carry the CAR receptor onto their surface to interact with the viral vector [14,15]. Nevertheless, although the strategy of using chimpanzee adenoviruses may reduce immune response failure following vaccination, the phase 2/3 trial with ChadOx1 nCoV-19 revealed that such pre-existing or acquired anti-vector immunity exists and it may reduce the efficacy of the vaccine since it negatively correlates with the anti-spike total IgG [16]. The models for evaluating the innate or acquired resistance towards adenoviruses viral vector mainly relies on models assessing secreted alkaline phosphatase, but they are not able to capture the whole complexity of the immunity response against the viral vector and its capacity to transfer the targeted gene.

In this case-report, we further investigate the case of an 83-year-old woman vaccinated with ChadOx1 nCoV-19 who developed a vaccine-induced thrombosis with thrombocytopenia syndrome (TTS) [17]. Interestingly, on top of her TTS, she did not develop an antibody response against the spike protein of SARS-CoV-2 following the administration of her first dose of ChadOx1 nCoV-19.

## 2. Experimental Section

### 2.1. Sample Collection and Handling

Blood samples were collected into serum-gel tubes (BD Vacutainer 8.5 mL tubes, Becton Dickinson, Franklin Lakes, NJ, USA) or lithium-heparin plasma tubes (BD Vacutainer 4.0 mL tubes, Becton Dickinson, Franklin Lakes, NJ, USA) according to standardized operating procedure and manufacturer recommendations. Two sera from the 83-year-old woman vaccinated with ChadOx1 nCoV-19 vaccine were collected on the 14th and the 15th day after vaccination. Five sera from 5 individuals doubly vaccinated with ChadOx1 nCoV-19 vaccine were collected on 14 July 2021. The timing of the collection is 30 to 35 days after the administration of the second dose.

### 2.2. Patient and Controls Description

A complete description of this case has been reported previously for the VITT event [17]. Briefly, an 83-year-old woman presented at the emergency room with an alteration of her general condition. She presented with symptoms of weakness, nausea, vomiting, weight loss and spontaneous bruises without any obvious reason, 14 days after having received her first dose of ChadOx1 nCov-19. Oxygen saturation was 98% at admission and the patient was tested negative for SARS-CoV-2 infection as assessed by reverse-transcriptase polymerase chain reaction (RT-PCR).

The diagnosis of VITT was confirmed using a heparin-induced multi-electrode aggregometry method and the measurement of anti-PF4 IgG antibodies [18]. In the face of the clinical picture, i.e., thrombocytopenia and thrombosis, with the presence of anti-PF4 antibodies and positive platelet activation tests within 30 days after vaccination with ChadOx1 nCov-19, VITT was diagnosed [19].

Unfortunately, the clinical status worsened on day 12 post-admission with a de novo reduction of platelet count, and she died on day 14 from cardiovascular collapse.

The controls used in this study were individuals having received two-doses of ChadOx1 nCoV-19 vaccine. This population included 4 women and 1 man. The median age was 24.5 years (min–max: 23–26 years).

### 2.3. Analytical Procedures

The response to the vaccine in control individuals was evaluated through the measurement of binding antibodies against the RBD of the S1 subunit of the SARS-CoV-2 spike protein using the Elecsys Anti-SARS-CoV-2 S assay that measured total antibodies (Roche Diagnostics) with a positivity cut-off of 0.8 BAU/mL. Additionally, total antibodies against the SARS-CoV-2 NCP (Roche Diagnostics) were measured using the Elecsys Anti-SARS-CoV-2 assay. Results above 0.165 cut-off index were considered positive [20]. These analyses were performed on a cobas 801.

A cellular model for assessing resistance to the ChadOx1 nCoV-19 in the Vaxzevria^®^ vaccine was developed. This test is based on the detection of the production of the spike protein (S protein) induced by ChadOx1 nCoV-19 (lot number: ABW4805) in the supernatant fraction of cells transfected by the adenovirus (Figure 1).

The model is then exposed to the serum of the tested patient. In presence of elements impairing the infection of the cells by ChadOx1 nCoV-19 as anti-adenovirus antibodies or other elements present in the serum to prevent the action of the adenovirus, the production of the S protein in the supernatant is reduced or abolished. Briefly, A549 cells (human lung adenocarcinoma cell line) were dispensed into a 24-well plate at an optimal concentration to achieve confluence without excessive cell death. This cell mat was then placed with a fixed amount of ChadOx1 nCoV-19 vaccine and a progressive dilution of the patient’s or control’s serum (dilutions ranging from 1/4 to 1/1024). The plate was then left for 7 days at 37 °C and 5% of CO_2_ in a calibrated incubator. Measurement of the amount of S protein present in the supernatant after 7 days of incubation was performed using an ELISA from Active Motif kit targeting total S protein (lot number: 38220000, Active Motif, Carlsbad, CA, USA). This kit enables measurement of protein S concentrations ranging from 2 ng/mL to 150 ng/mL. The absorbance of the test solution was carried out on the Spectramax 3ID (Molecular Devices, San Jose, CA, USA). The summary of the protocol is shown in Figure 2.

### 2.4. Statistical Analysis

Mean and 95% confidence intervals (95% CI) were used for descriptive statistics. A data analysis was performed using GraphPad prism software (version 8.2.1, San Diego, CA, USA).

## 3. Results

The controls all have positive anti-SARS-CoV-2 S protein antibodies titers with a mean titer of 2427 AU/mL (95% CI: 1581 AU/mL–6434 AU/mL) and negative anti-NCP antibodies titers. The results for the serum analysis of the patient not responding to Vaxzevria^®^ are presented in Table 1.

The absorbance does not vary according to the serum dilution and remains between 0.06 and 0.08. These values were below the limit of quantification (LOQ) of the ELISA assay (LLOQ = 2 ng/mL) and did not allow the calculation of a protein S concentration in the supernatant. The results obtained with the controls provide mean concentrations for the different serum dilutions ranging from 21.60 ng/mL (95% CI: 17.20 ng/mL–26.00 ng/mL) to 24.52 ng/mL (95% CI: 17.23 ng/mL–31.81 ng/mL) (Table 2). The overall mean for the Vaxzevria^®^ control group is 23.10 ng/mL (95% CI: 22.31 ng/mL–23.89 ng/mL).

## 4. Discussion and Conclusions

It has been demonstrated that A549 cells are highly transfectable cells for adenoviruses such as ChadOx nCoV-19, which is in line with other studies such as those of Zhong et al. [21,22]. Based on this information, and on the results obtained, it can be assumed that the clinical case presented in this paper developed a form of immunity against the adenovirus used in the ChadOx1 nCoV-19 vaccine. Indeed, when her serum was used in the vaccine cell model, an absence of S protein production was observed, which is not the case in controls. The origin of this immunity is still unknown, but this test allows to eliminate a possible lymphocytic or myelocytic origin. In the case of a failed vaccine response due to lymphocyte or bone marrow disorders, the test presented would have shown no inhibition of protein S production because the serum in this case would not have contained any anti-ChAdOx1 nCoV-19 antibodies. It is not new to hear about resistance against adenovirus therapies as theses therapies have been used for years in the treatment of certain types of cancer. It is in this context that Zhang L. et al. reported in 2002 the development of resistance to a form of cancer therapy involving adenoviruses. In their case, the resistance to the therapy is due to a resistance to the infection of the cells by the adenovirus used [23]. Liikanen I. et al. also reported in 2011 the presence of resistance to therapies using adenoviruses as vectors. They highlight several resistance pathways involving ribonuclease L or protein kinase R and others interferon pathways [24]. Another hypothesis is that the patient may have antibodies directed not against the adenovirus used in the Vaxzevria^®^ vaccine but against her own CAR receptor. This would explain the impossibility of interaction between this receptor and the adenovirus and therefore the lack of response to the vaccination.

In this case report, we have demonstrated the presence of resistance to the Vaxzevria^®^ vaccine in a patient who developed TTS. The origin of this resistance could not be determined but several hypotheses were formulated. Further development of the in-house model and additional testing would be required to identify the precise cause of the patient’s immunity. Unfortunately, the failure of large-scale vaccination with the Vaxzevria^®^ vaccine in our country has led to the disappearance of this product from our market, making further testing difficult. In addition, expanding the control group or the number of patients is also difficult to envisage given the small number of people vaccinated with only Vaxzevria^®^ vaccine. The model developed is applicable to other therapies using adenoviruses such as anti-cancer therapies or several vaccines already on the market such as Ebola vaccine (Zabdeno^®^) or other COVID-19 vaccine as the Johnson & Johnson vaccine (Jcovden^®^). In addition to these therapies already on the market, several studies are underway to develop a malaria vaccine based on adenoviruses such as Ad35 or Ad26 [25,26]. There are also studies on a booster that would follow this vaccine based on adenovirus such as adenovirus-associated virus-serotype 1 (AAV1) or AAV8 [27,28]. The test developed would therefore make it possible to assess an individual’s resistance to an adenovirus-based therapy more widely. This would make it possible to prevent the use of certain therapies that we know will not work in a particular individual. Importantly, a link with the TTS developed by our patient cannot be excluded and deserved further investigations.

Finally, there are several limitations to the study. First, the difficulty in obtaining VAXZEVRIA vaccine makes the manipulation impossible to replicate in some countries that no longer use this vaccine. However, the model presented remains a basis for investigation of other adenovirus vaccines. The second main limitation is the low number of controls, due to the small proportion of the population vaccinated with VAXZEVRIA alone.

## Figures and Tables

**Figure 1 vaccines-10-02056-f001:**
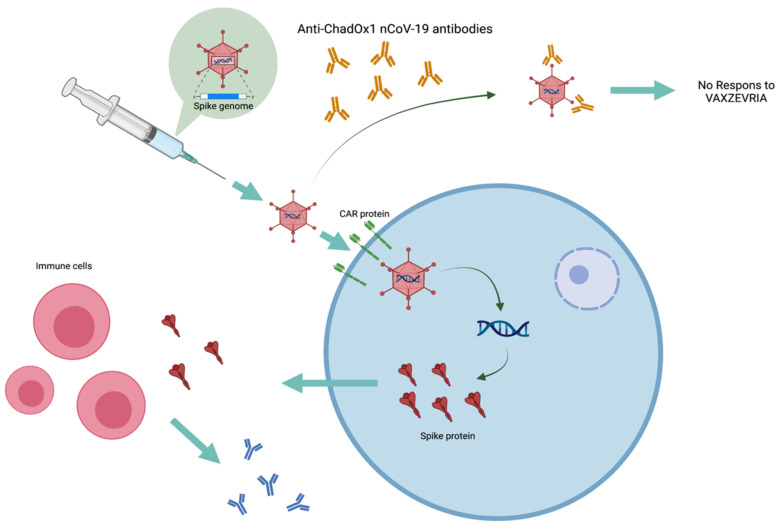
The schema of the mechanism of action of the Vaxzevria^®^ vaccine and the consequences of prior immunization against ChadOx1 nCoV-19. In this figure the different stages of the response to Vaxzevria^®^ vaccine can be distinguished. This response starts with the injection and interaction of ChADOx1 nCoV-19 with cells expressing the CAR protein. This is followed by an intracellular process leading to the production and secretion of protein S. This S-protein outside the cells will be recognized by the immune system allowing the production of anti-S antibodies. In the case of pre-existing immunity to ChADOx1 nCoV-19, the antibodies present will prevent interaction with cells expressing the CAR protein, leading to a lack of response to the Vaxzevria^®^ vaccine.

**Figure 2 vaccines-10-02056-f002:**
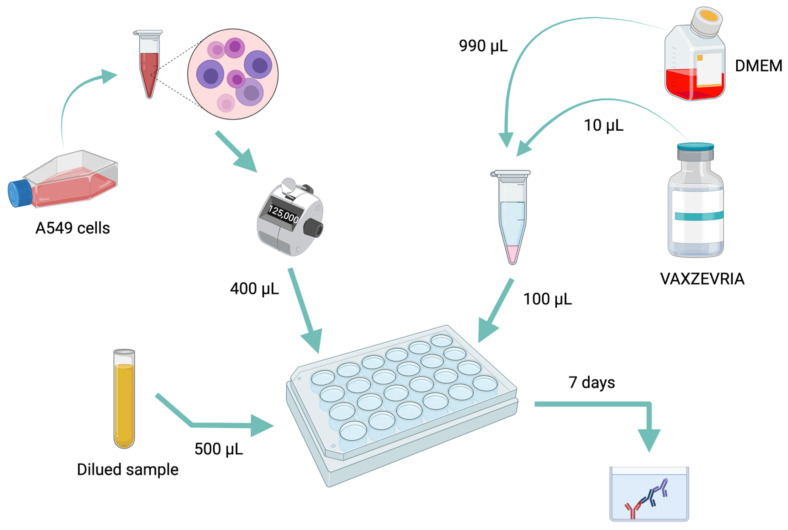
A summary diagram of the handling protocol.

**Table 1 vaccines-10-02056-t001:** The absorbance result obtained at D0 and D1 for the patient according to the serum dilutions.

Sample Dilution Factor	Sample D0 Absorbance	Sample D1 Absorbance
1/4	0.07	0.07
1/8	0.07	0.07
1/16	0.06	0.08
1/32	0.06	0.08
1/64	0.06	0.07
1/128	0.08	0.06
1/256	0.07	0.06
1/512	0.08	0.08
1/1024	0.07	0.07

**Table 2 vaccines-10-02056-t002:** The concentration of protein S in the supernatant obtained for the different dilutions of the Vaxzevria^®^ controls sera.

Sample Dilution Factor	VAXZEVRIA Double-Vaccinated Patients Mean Concentration (ng/mL)
1/4	24.52 (95% CI: 17.23–31.81)
1/8	23.36 (95% CI: 17.00–29.71)
1/16	24.28 (95% CI: 17.21–31.35)
1/32	21.98 (95% CI: 14.82–29.13)
1/64	22.71 (95% CI: 14.24–31.19)
1/128	22.81 (95% CI: 16.15–28.99)
1/256	22.57 (95% CI: 16.15–28.99)
1/512	21.60 (95% CI: 17.20–26.00)
1/1024	24.06 (95% CI: 18.16–29.96)
**Overall mean** **(ng/mL)**	23.10 (95% CI: 22.31–23.89)

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
