# Peer review of "Resistance towards ChadOx1 nCoV-19 in an 83 Years Old Woman Experiencing Vaccine Induced Thrombosis with Thrombocytopenia Syndrome"

_vaccines, 2022, doi:10.3390/vaccines10122056_

Round 1

Reviewer 1 Report

This is an important report showing the lack of production of the spike protein after adding the serum from a patient who developed thrombosis with thrombocytopenia syndrome (TTS) after the ChAdOx1 nCoV-19 coronavirus vaccine (recombinant). The methodology section is well described with illustrations. The in vitro study is not an evidence linking between lack of S protein production and the development of TTS, as the authors indicated. Descriptions in the discussion & conclusion sections are also balanced for the pros and cons of the in vitro study in this report.

Minor

1) Days after vaccination should be described for control sera if the information is available (Line 103–104).

2) The minus before ‘1581 AU/mL’ in Line 178 might be a typo.

Author Response

Dear Reviewer 1,

  • Days after vaccination should be described for control sera if the information is available (Line 103–104).

We thank you for your comment and base and your remark, we added “The timing of the collection is 30 to 35 days after the administration of the second dose.” (Line 128 - 129)

  • The minus before ‘1581 AU/mL’ in Line 178 might be a typo.

We thank you for your comment and base and your remark, we corrected this typo. (Line 220)

.

Yours sincerely,

Constant Gillot,

On behalf of the authors of this study.

Reviewer 2 Report

This manuscript by Gillot et al. is interesting, but I have some concerns about it.

At first,it is difficult to clearly understand if it is a case report or a proof of concept study. In fact, the authors stated that the reported case has been published in Ref. #17, and in such case this one is a duplicate publication. If not, I suggest to refer to the previous study deleting the extensive description of the case, focusing on the proposed study methods.

However, the authors report that in their country the evaluated vaccine and the similar one from another manufacturer are no more available, as well as in several countries worldwide. As a consequence, the study is very difficult (or impossible) to replicate.

Limitations of the study, including the last above reported, are not defined.

Line 117: the use of NYHA scale for dyspnea is misleading and confusing. I suggest to use another, specific evaluation scale for dyspnea (i.e., MRC).

References must be revised according to the journal style.

I also suggest English grammar and language revision of the manuscript.

Author Response

Dear Reviewer 2,

  • At first,it is difficult to clearly understand if it is a case report or a proof of concept study. In fact, the authors stated that the reported case has been published in Ref. #17, and in such case this one is a duplicate publication. If not, I suggest to refer to the previous study deleting the extensive description of the case, focusing on the proposed study methods.

We thank you for your remark, according to your comment, we have reduced the description of the case as it is described in the cited reference. However, for us it is not a question of a duplicate publication but rather of additional investigations on the same clinical case.

  • However, the authors report that in their country the evaluated vaccine and the similar one from another manufacturer are no more available, as well as in several countries worldwide. As a consequence, the study is very difficult (or impossible) to replicate.

We thank you for your comment, we are aware of this difficulty in reproducing the manipulation in some countries of the world, but there are still countries using the vaccine marketed by AstraZeneca.

Furthermore, the model presented could serve as a basis for investigation of other adenovirus-based vaccines.

  • Limitations of the study, including the last above reported, are not defined.

We thank you for your comment, we added some limitations at the end of the discussion. (Line 276 - 281)

  • Line 117: the use of NYHA scale for dyspnea is misleading and confusing. I suggest to use another, specific evaluation scale for dyspnea (i.e., MRC).

We thank you for your comment, the section mentioned is no longer present in the case description, which has been shortened.

  • References must be revised according to the journal style.

We thank you for your comment, reference style has been modified.

Yours sincerely,

Constant Gillot,

On behalf of the authors of this study.

Round 2

Reviewer 2 Report

The authors provided acceptable replies and improved the manuscript